# Are Metal-Free Monolithic Crowns the Present of Prosthesis? Study of Mechanical Behaviour

**DOI:** 10.3390/ma12223663

**Published:** 2019-11-07

**Authors:** Rubén Agustín-Panadero, Raquel León Martínez, María Fernanda Solá-Ruíz, Antonio Fons-Font, Georgina García Engra, Lucía Fernández-Estevan

**Affiliations:** 1Department of Dental Medicine, Faculty of Medicine and Dentistry, University of Valencia, 46010 Valencia, Spain; m.fernanda.sola@uv.es (M.F.S.-R.); antonio.fons@uv.es (A.F.-F.); georgina.g.engra@gmail.com (G.G.E.); lucia.fernandez-estevan@uv.es (L.F.-E.); 2Private Practice, 46008 Valencia, Spain; raquelleonmtnz@gmail.com

**Keywords:** aging, dental crown, fracture resistance, graphene, monolithic zirconia

## Abstract

Purpose: To analyze in vitro the mechanical behavior of five types of complete coverage crowns fabricated from different materials. Materials and methods: Seventy-five full coverage crowns were divided into five groups according to material: Group I, metal core with feldspathic ceramic covering (MC- control group); Group II, zirconia core with feldspathic ceramic covering (CZ); Group III, tetragonal monolithic zirconia (TMZ); Group IV, cubic monolithic zirconia (CMZ); Group V, high molecular weight polymethyl methacrylate (PMMAG) doped with graphene nanoparticles. All crowns underwent in vitro fatiguing by dynamic loading in wet conditions to simulate the masticatory forces to which prosthodontic materials are subject in the oral medium. Lastly, fracture resistance was evaluated by static compression testing. Results: The fracture resistance values obtained were as follows: Group MC, 2443.6 ± 238.6 N; Group CZ, 2095.4 ± 329.2 N; Group TMZ, 2494.6 ± 236.6 N; Group CMZ, 1523.6 ± 325.2 N; and Group PMMAG, 1708.9 ± 386.6 N. Group MC presented higher strength with statistically significant differences in comparison with Groups CZ (*P* = 0.002), CMZ (*P* < 0.001), and PMMAG (*P* < 0.001). Weibull distribution showed less probability of cumulative biomechanical failure in Groups MC and TMZ. Conclusions: Metal-ceramic and tetragonal zirconia showed high fracture resistance, while cubic zirconia and PMMA doped with graphene nanoparticles obtained lower values.

## 1. Introduction

The rehabilitation of lost dental tissue is one of the main objectives of prosthetic dentistry. Full coverage crowns are the classic therapeutic option for restoring teeth that have suffered major destruction. The materials used to fabricate these prothesis must provide mechanical properties that guarantee their clinical success and survival. When new materials are introduced into the field of prosthetic dentistry, one of the objectives is to ensure that crowns or fixed partial dentures provide adequate resistance to mastication and a level of endurance greater or similar to the classic materials [1].

The strength of a prosthodontic restoration is largely determined by its composition. The metal–ceramic crowns are the most studied in the literature because they have been used for many years. However, the metal-free restorations are developed more popular in last years for supply the disadvantages in its esthetic appearance and biocompatibility [1,2,3,4,5]. In this context, porcelain-veneered crowns with zirconia cores have been introduced and are supported by some studies that confirm the good resistance of this material [2,6,7,8], although its main disadvantage is caused by a cohesive failure between both components and is known in the literature as chipping is seen in 10%–15% of crowns [6,7,8,9,10,11]. This problem has generated some doubt as to the prognosis of these materials in the long term. To avoid this clinical complication, monolithic zirconia crowns were developed in an attempt to achieve the greatest possible strength in a metal-free restoration and eliminate the chipping problem. Yttrium-stabilized tetragonal zirconia obtains a fracture resistance similar to metal-ceramic crowns, but its aesthetics are compromised by the high opacity of the blocks the crowns are fabricated from [6,7,8]. In recent years, yttrium-stabilized cubic zirconia has been introduced as it offers more translucency but at the expense of mechanical properties (strength and hardness) [12,13].

One of the advantages of monolithic zirconia crowns is that they require less dental preparation, with reductions of 0.5 mm compared with the 1.2–1.5 mm needed to place crowns with zirconia cores and porcelain coverings [14]. This means removing about 50% less dental tissue and so constitutes a much more conservative treatment. Another recent development has been the introduction of monolithic materials based on polymethyl methacrylate (PMMA) doped with graphene nanoparticles that have shown promising properties and perhaps represent the future of restorative dentistry, although to date their properties have not been evidenced by in vitro or in vivo trials [15]. Monolithics materials are made with CAD/CAM techniques that require only a few steps and less time for the fabrication of a restoration compared to old traditional methods, but the literature shows survival rates slightly lower than the restorations with a metal/zirconia core and porcelain veneer [16,17]. CAD/CAM systems also have some disadvantages. The scanning system has the limitation of finite resolution, which can result in edges that are slightly rounded. The point clouds obtained in scanning are transformed through a CAD software into a smooth and continuous surface, which in some cases can cause reproducibility errors of the anatomy of the preparation that will influence the marginal and internal fit of the prosthetic restoration [16]. The aim of this trial was to analyze the in vitro fracture resistance of metal-free zirconia and polymer-based crowns (TMZ, CMZ, PMMAG) compared with conventional crowns (metal/zirconia core with ceramic covering, MC-CZ).

## 2. Material and Methods

The crowns used in this trial were designed from a resin tooth model with the shape of a maxillary first molar. Dental preparation was performed following the principles recommended by Schillingburg for full coverage crowns with a functional cusp occlusal reduction of 1.5–2 mm and non-functional cusp reduction of 1 mm, axial wall reduction of 1.2–1.5 mm, and 6° convergence of the axial walls [18]. Using a copper cylinder as an impression tray, an impression of the tooth stump was taken with double addition silicone of heavy (Putty Elite HD, Zhermack, Badia Polesine, Italy) and light consistency (Light Elite HD, Zhermack). Afterwards, 75 specimens were cast in epoxy resin (Exakto-Form, Bredent, Send, Germany) to obtain replicas of a prepared tooth. The specimens were set in 22 mm diameter copper cylinders using the same resin. 

Having made a wax-up previously, the morphology of the test crowns was designed to create three-point (tripod) occlusal contact between the internal slopes of the vestibular and palatine cusps on the restoration and the test machine’s antagonist load applicator (a 4 mm diameter aluminum ball). Both the wax-up and the tooth stump were scanned digitally with a True Definition intraoral scanner (3M ESPE, Maplewood, MN, USA) to generate two STL files. These were used to design the crowns digitally with Exocad software (Exocad DentalCAD version 5164, ExocadGmbH, Darmstadt, Germany) (Figure 1). Seventy-five crowns were divided into five groups according to their fabrication materials: Group MC (n = 15) metal-ceramic (Cr-Co internal core (Rexilium V, Argen, San Diego, CA, USA) + stratified feldspathic ceramic covering (IPS d.SIGN, Ivoclar Vivadent, Schaan, Liechtenstein) (control); Group CZ (n = 15) zirconia internal core (Lava Frame Zirconia, 3M ESPE) + stratified feldspathic ceramic covering (Lava Ceram, 3M ESPE); Group TMZ (n = 15) 3 mol% yttrium-stabilized tetragonal zirconia (Lava Plus, 3M ESPE); Group CMZ (n = 15) 5 mol% yttrium-stabilized cubic zirconia (Lava Esthetic, 3M ESPE); Group PMMAG (n = 15) PMMA doped with graphene nanoparticles (Dental Graphene). The crowns were cemented onto the epoxy resin stumps with a self-cure composite resin cement (Rely X Unicem 2 Automix, 3M ESPE) applying a 1 kg load for the first 5 min setting time. 

Each specimen underwent dynamic compression fatiguing (Chewing Simulator CS-4 SD Mechathronik, Lava Plus) for 60,000 cycles in wet conditions (saline) (Figure 2A). The load applied was 80N with 2 mm displacement, protrusive displacement of 2 mm, and 2 Hz frequency. Afterwards, specimens underwent compression testing using a universal test machine (AGS-X Shimadzu, Kyoto, Japan) with a 5000 N load cell and a crosshead speed of 0.5 mm/min (Figure 2B) until the point of fracture. The fracture resistance data obtained were analyzed calculating descriptive statistics, and probability of failure was estimated by means of the Weibull modulus and scale parameter (m). The confidence interval was 95% and the power to detect differences between mean resistance values was 0.77.

## 3. Results

The mean fracture resistance values obtained in compression testing were as follows: Group MC, 2443.6 ± 238.6 N; Group CZ, 2095.4 ± 329.2 N; Group TMZ, 2494.6 ± 236.6 N; Group CMZ, 1523.6 ± 325.2 N; and Group PMMAG, 1708.9 ± 386.6 N (Table 1) (Figure 3). Statistically significant differences between mean values were found, whereby Group MC (control group, metal-ceramic) showed significantly higher resistance to fracture than Groups CZ (ceramic-zirconia) (*P* < 0.005; Bonferroni test), CMZ (cubic monolithic zirconia) (*P* < 0.001; Bonferroni test), and PMMAG (PMMA with graphene) (*P* < 0.001; Bonferroni test). 

Group TMZ showed significant differences in comparison with Group CZ (*P* < 0.001; Bonferroni test), Group CMZ (*P* < 0.001; Bonferroni test), and Group PMMAG (*P* < 0.001; Bonferroni test) (Figure 4). 

Comparisons of the fracture resistance of non-monolithic (Groups MC and CZ) with monolithic crowns (Groups TMZ, CMZ and PMMAG) show statistically significant differences (*P* = 0.001; T Test), non-monolithic crowns (Groups MC and CZ) being stronger (Figure 5). 

Analyzing the internal core materials of the crowns (Figure 6): metal (Group MC), zirconia (Groups CZ, TMZ and CMZ), and PMMA with graphene nanoparticles (Group PMMAG), metal core crowns (Group MC) were found to be significantly more resistant to fracture than zirconia core crowns (Group CZ) (*P* < 0.001; Tanhane test) and graphene-doped PMMA (Group PMMAG) (*P* < 0.001; Tanhane test). Zirconia core crowns (Group CZ) were also significantly more resistant to fracture than graphene-doped PMMA (Group PMMAG) (*P* = 0.038; Tanhane test). When monolithic zirconia crowns were compared according to the crystallographic phase at which they had been stabilized, tetragonal zirconia (Group TMZ) presented significantly higher fracture resistance than cubic zirconia (Group CMZ) (the latter offers better esthetics but less strength) (*P* < 0.001; T-test).

Calculating Weibull distribution, Group MC (control) (m = 12.04) and Group TMZ (m = 12.72) obtained higher distributions, which indicates more predictable behavior. Graph represents the higher probability scale of these two groups, similar to one another but higher than Groups CZ, CMZ, and PMMAG, which obtained lower distributions, with Groups CMZ and PMMAG showing the lowest distributions (m = 5.45 and 5.02, respectively) (Figure 7).

## 4. Discussion

The literature presents an obvious scarcity of in vitro clinical trials of metal-free crowns, particularly trials of those materials that claim to introduce innovative advantages to the field of prosthetic dentistry. The present study used metal-ceramic full coverage crowns as its control group, as according to the literature is the most supported [1,2,3,4,5,10]. The trial compared the control crowns with crowns made from other monolithic and non-monolithic materials with ceramic coverings [2,3,4,5,6,7,8,11], including PMMA doped with graphene nanoparticles, a recently launched dental material [15]. Monolithic zirconia crowns are made with CAD/CAM techniques and allow more conservative dental preparation [13,16,17]. In addition, the thickness allows us to obtain higher translucency values and a decreased contrast ratio to improve a long-term aesthetic compared with metal-ceramic [18]. Monolithic zirconia crowns may be stabilized in tetragonal phase, when they present a strength equal to metal-ceramic crowns but with the disadvantage of high opacity [2,3,4,8]. For this reason, zirconia crowns stabilized in cubic phase have been introduced, which offer greater translucency and so better aesthetics, but less strength and hardness [12,13]. Camposilvan analyzed the hardness of zirconia stabilized with different proportions of cubic structure and found that as the percentage of cubic zirconia increased so did its translucency, while strength and hardness decreased [13].

The specimens fabricated in the present in vitro trial corresponded to the anatomical form of a maxillary first molar in order to equal the scale of a real tooth. The occlusal morphology of the crowns was determined by means of a wax-up seeking three-point (tripod) contact between the inner slopes of the vestibular and palatine cusps and the test machine’s load applicator, as in previous trials conducted by Agustín, López-Suarez, and Bindl [2,3,14]. Other authors have performed compression testing with samples in the form of discs or cylinders, while the present work, in agreement with Mori, fabricated crowns with homogenous internal cores (Groups MC and CZ) so that porcelain coverings would be of uniform thickness [10].

The compression testing used in the present trial has been described as the most effective means of evaluating the fracture resistance of crowns and bridges and has been used by various authors in the literature. It also fulfills the requirements and recommended methods established in ISO 6872:2015 directives. Dynamic fatiguing consisted of 60,000 cycles of cyclic loading in wet conditions (saline). The load applied was 80 N with vertical displacement of 2 mm and protrusive displacement of 2 mm and 2 Hz frequency. In static load testing, a load cell of 5000 N was used at a crosshead speed of 0.5 mm/min until fracture, a procedure used in many other similar trials [2,3,8,11,12].

Weibull distribution analyzes a structure’s variable failure rate. When assessing ceramic materials, Weibull statistical analysis expresses the probability of failure and so greater understanding of a material’s biomechanical behavior [19,20,21]. Groups MC and TMZ obtained higher scales, being materials with greater long term reliability. This finding also corresponds to the higher fracture resistance found in compression testing. Groups CZ, CMZ, and PMMAG obtained the lowest Weibull scale parameter values and so less predictable long-term biomechanical behavior. Regarding the results obtained with the newer materials investigated, the data obtained support the use of tetragonal monolithic zirconia (Group TMZ) for fabricating full coverage crowns, while cubic monolithic zirconia (Group CMZ) and graphene-doped PMMA (Group PMMAG) showed worse prognoses. 

As for the mean mastication force exerted in the posterior region (700 N rising to a maximum of 860–900 N) [22] it was affirmed that all the crowns tested presented much higher fracture resistance values (Group MC, 2443.6 N; Group CZ, 2095.4 N; Group TMZ, 2494.6 N; Group CMZ, 1523.6 N; Group PMMAG, 1708.9 N). 

Metal-ceramic crowns obtained a mean fracture resistance of 2443.675 N, a similar value to results published by López-Suarez (3008.69 N), Sun (2284.7 N), and Agustín (2310.49 N) [2,3,4]. However, for zirconia core crowns with ceramic covering, the literature shows greater heterogeneity of fracture resistance values, although Bindl, Agustín, and Tsalouchou have obtained similar values to the present trial—1896.5 N, 1992.45 N, and 2185.6 N respectively [3,7,14].

In a study by Sun, monolithic zirconia crowns with an occlusal thickness of 1 mm showed similar fracture resistance values to metal-ceramic crowns without statistically significant difference [4]. In the present trial, these two groups obtained similar values: Group MC, 2443.675 N and Group TMZ, 2494.649 N. 

Like other studies such as Camposilvan and Lawson, the present trial found significant differences between tetragonal monolithic zirconia (Group TMZ) and cubic monolithic zirconia (Group CMZ), with higher fracture resistance obtained in Group TMZ [12,13]. 

To date, no in vitro research into crowns fabricated from graphene-doped PMMA has been published, although on the basis of the present findings, it can be affirmed that the material reached fracture resistance values that exceeded maximum mastication forces, although the specimens presented a certain heterogeneity of results pointing to a need for further research before adequate clinical performance can be guaranteed. 

Tetragonal monolithic zirconia full coverage crowns are an advisable restoration option as they present a mechanical performance equal to conventional metal-ceramic crowns with the added advantage of better aesthetics and more conservative dental preparation. 

## 5. Conclusions

Yttrium-stabilized tetragonal zirconia is the only other material used to fabricate full coverage crowns that equaled the fracture resistance obtained by the metal-ceramic control group. Following these two materials in decreasing order of fracture resistance values were zirconia core with feldspathic ceramic covering, PMMA doped with graphene nanoparticles, and yttrium-stabilized cubic zirconia. Within the monolithic zirconia crowns tested, tetragonal zirconia presented higher fracture resistance than cubic zirconia. All the crowns analyzed obtained adequate load resistance values for correct oral behavior. Prospective medium and long-term clinical studies are needed to support the survival of restorations made with CAD/CAM monolithic materials. 

## Figures and Tables

**Figure 1 materials-12-03663-f001:**
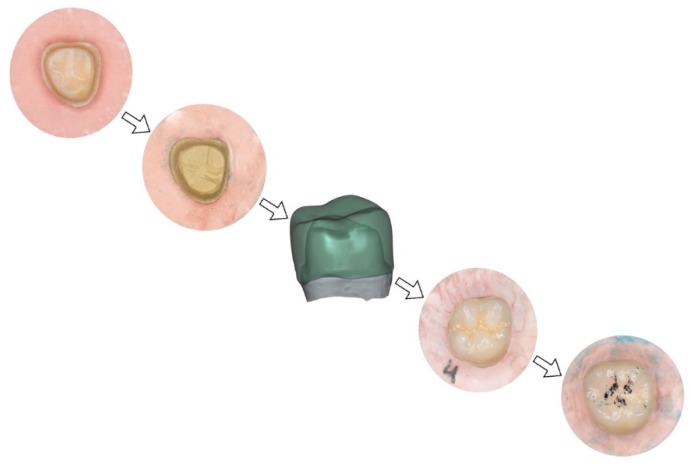
Process of design samples.

**Figure 2 materials-12-03663-f002:**
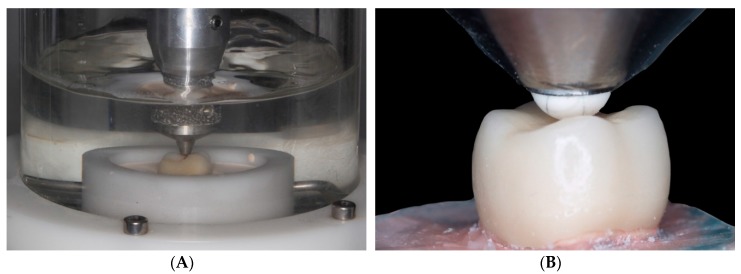
In vitro fracture resistance testing of crowns. (**A**) Dynamic compression fatiguing (Chewing Simulator CS-4 SD Mechathronik); (**B**) Static compression testing until fracture with SHIMADZU^®^ universal test machine.

**Figure 3 materials-12-03663-f003:**
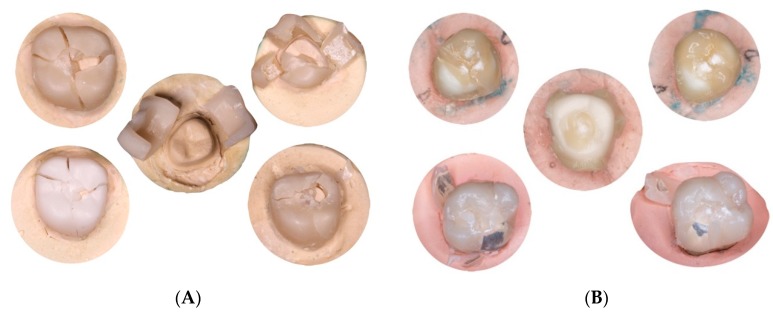
Crowns fractured after compression testing. (**A**) Monolithic crowns; (**B**) non-monolithic crowns.

**Figure 4 materials-12-03663-f004:**
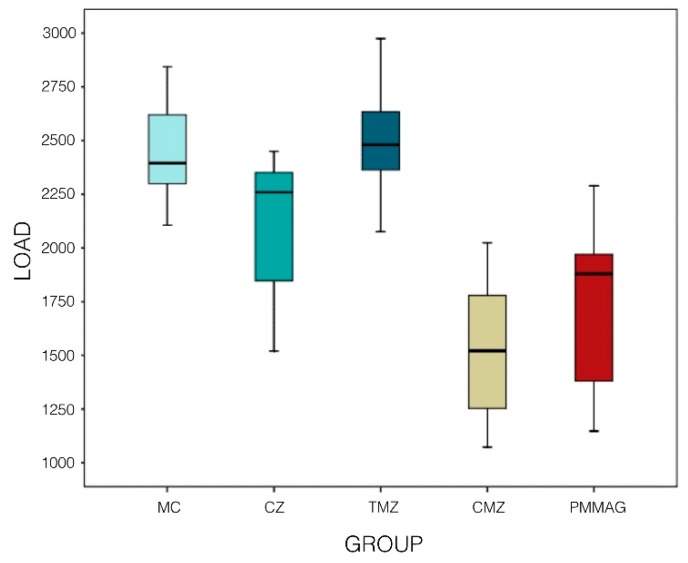
Box-plot shows fracture resistance in different groups.

**Figure 5 materials-12-03663-f005:**
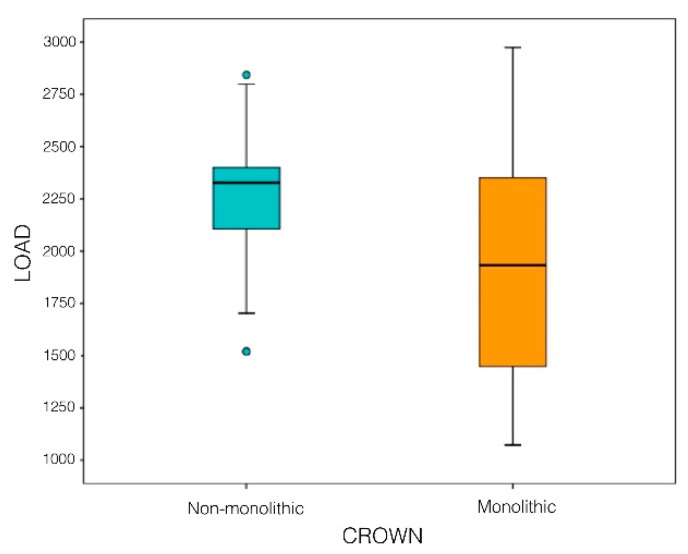
Box-plot shows fracture resistance of non-monolithic (Groups I and II) compared with monolithic crowns (Groups III, IV, and V).

**Figure 6 materials-12-03663-f006:**
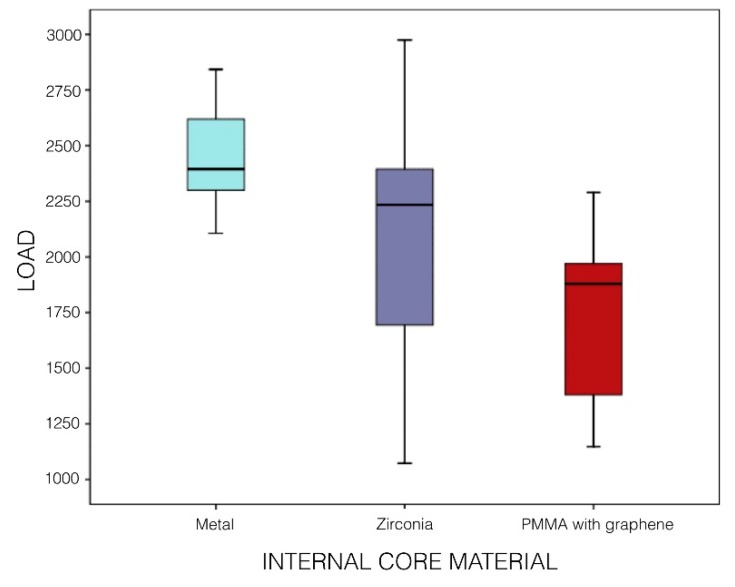
Box-plot shows comparison of fracture resistance between groups according to core material: metal (Group I); zirconia (Groups II, III, and IV); graphene-doped PMMA (Group V).

**Figure 7 materials-12-03663-f007:**
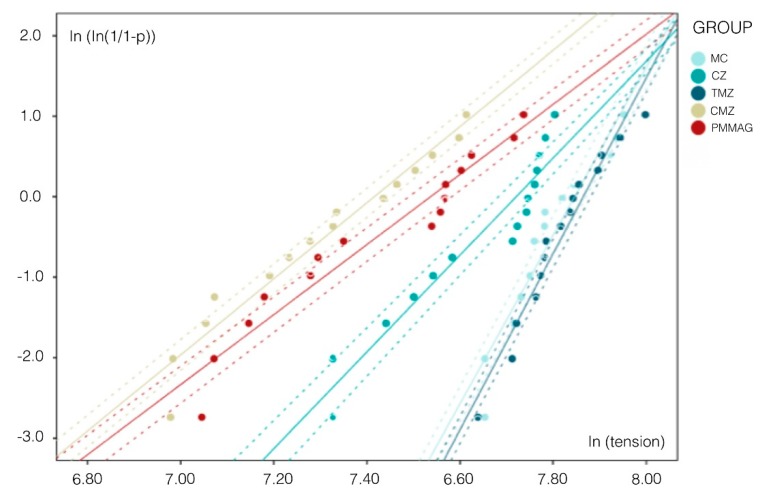
Graph showing Weibull Modulus for Groups I, II, III, IV, and V and 95% confidence interval calculated.

**Table 1 materials-12-03663-t001:** Resistance results of each group after the dynamic and static test.

Specimens	Metal-Ceramic	Zirconia-Ceramic	Tetragonal Monolithic Zirconia	Cubic Monolithic Zirconia	Polymethyl Methacrylate (PMMA) Doped with Graphene Nanoparticles
1	2685.87 N	2236.254 N	2394.835 N	1384.274 N	1449.108 N
2	2488.602 N	2399.993 N	2688.01 N	1882.76 N	1312.192 N
3	2842.947 N	2345.123 N	2376.143 N	1532.125 N	2289.915 N
4	2395.1 N	1808.736 N	2532.037 N	1179.266 N	1937.071 N
5	2553.012 N	2259.367 N	2234.173 N	1072.836 N	1555.745 N
6	2106.848 N	1964.957 N	2547.455 N	1520.888 N	1147.493 N
7	2332.87 N	2310.984 N	2255.185 N	1157.506 N	2002.78 N
8	2277.845 N	1885.435 N	2579.959 N	1744.477 N	1472.441 N
9	2761.99 N	1520.135 N	2075.45 N	1447.455 N	2242.947 N
10	2397.889 N	2365.786 N	2480.427 N	1993.116 N	1177.994 N
11	2106.443 N	2304.045 N	2404.181 N	1694.028 N	1933.257 N
12	2798.884 N	2449.68 N	2974.749 N	1327.419 N	1879.851 N
13	2240.876 N	1704.029 N	2818.966 N	1078.637 N	1915.503 N
14	2342.765 N	1520.135 N	2350.442 N	1814.381 N	2047.491 N
15	2321.767 N	2356.276 N	2707.736 N	2024.667 N	1269.293 N
**Mean**	2443.58 N	2095.39 N	2494.64 N	1523.58 N	1708.87 N

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
