# Peer review of "Are Metal-Free Monolithic Crowns the Present of Prosthesis? Study of Mechanical Behaviour"

_materials, 2019, doi:10.3390/ma12223663_

Round 1

Reviewer 1 Report

Dear authors, 

even though the methods is well described, 

I found the introduction not precisely exposed. 

Usually in dentistry the terms restorative is used for small cavities, and crowns for the prosthodontics. You have to take a decision. 

the sentences are both in intro and in the discussion are not well referenced

in particular saying the gold standard is still the metal-ceramic crowns is really old fashioned. 

In my opinion the intro and discussion should be more improved to make tha paper worthy of reading, 

Author Response

I agree with you that crowns are for prosthetic dentistry in despite of restorative dentistry, and this has been an incorrect translation of Spanish to English. We changed the sentence:

Line 31: is one of the main objectives of prosthetic dentistry.

Line 33: changed ‘restoration’ for ‘prothesis’.

Line 35: changed restorative dentistry for ‘prosthetic dentistry‘ and ‘restoration’ for ‘crowns or fixed partial dentures’.

Line 150-1: to the field of prosthetic dentistry.

Line 37: deleted considered ‘standard’.

We have changed the narration and exposure of the introductionand we have added three recent articles to make it easier to understand it and more interesting to the reader:

The strength of a prosthodontic restoration is largely determined by its composition. The metal-ceramic crowns are the most studied for the literature because has been used for many years. But the metal-free restorations are developed more popular in last years for supply the disadvantages in its esthetic appearance and biocompatibility [1-5].In this context, porcelain-veneered crowns with zirconia cores have been introduced and are supported by some studies that confirm the good resistance of this material [2,6-8], although its main disadvantage is caused by a cohesive failure between both components and its known in the literature as chipping seen in 10-15% of crowns [6-11]. This problem has generated some doubt as to the prognosis of these materials in the long term. To avoid this clinical complication, monolithiczirconiacrowns were developed in an attempt to achieve the greatest possible strength in a metal-free restoration and eliminate the chipping problem.

Monolithics materials are made with CAD/CAM techniques that requires only few steps and less time for the fabrication of a restoration compared to old traditional methodsbut the literature shows survival rates slightly lower than the restorations with core (metal / zirconia) and porcelain-veneered [16,17].CAD/CAM systems also have some disadvantages. The scanning system has the limitation of finite resolution, which can result in edges that are slightly rounded. The point clouds obtained in scanning are transformed through a CAD software into a smooth and continuous surface, which can also lead to some internal that will influence the marginal and internal fit of the prosthetic restoration [16].

Line 152: In discussion, we changed ‘gold standar’ to as according to the literature is the most supported, in order to justify the use of metal-ceramic like control group.

We introduced some new sentences in discussion:

Monolithic  zirconia crowns are made with CAD/CAM techniques and allow more conservative dental preparation [13,16,17]. In addition, the thickness allows us to obtain higher translucency values and a decreased contrast ratio to improve a long-term aesthetic compared with metal-ceramic. 

We have modified the conclusions to adapt them to the results obtained

Yttrium-stabilized tetragonal zirconia is the only other material used to fabricate full coverage crowns that equaled the fracture resistance obtained by the metal-ceramic control group.Following these two materials in decreasing order of fracture resistance values were: zirconia core with feldspathic ceramic covering, PMMA doped with graphene nanoparticles, and yttrium-stabilized cubic zirconia. Within the monolithic zirconia crowns tested, tetragonal zirconia presented higher fracture resistance than cubic zirconia. All the crowns analyzed, internal core and monolithic restorations, obtain adequate load resistance values for correct oral behavior. Prospective medium and long-term clinical studies are needed to support the survival of restorations made with CAD/CAM monolithic materials. 

Reviewer 2 Report

the research is interesting and well reported. it should be interesting to have information about the crowns used as restoration in implantology. Can the results of this research be used also in implant dentistry?  

Author Response

Reply:Thank you very much for your comments

In this case, the ‘in vitro’ study is made with specimens of epoxy resin similar to teeth and the results of resistance are for the crowns thinking in tooth-soported and we preferred to focus our results on crowns for natural teeth. But it is true that another ‘in vitro’ studies can be carried out with the same materials cemented on implant abutments.

Reviewer 3 Report

This is a very interesting and valuable publication in the field of prosthodontics and dental materials. The study was very well designed and performed as well as the results were correctly interpreted. The work provides a valuable guidance for clinical practice. The quality of figures 4 and 5 needs improvement. In the introduction, authors should mention popular CAD/CAM technology in dental restorations manufacturing. Moreover, authors should indicate the disadvantages of such technology (e.g. Rodrigues SB et al.: CAD/CAM or conventional ceramic materials restorations longevity: a systematic review and meta-analysis. J Prosthodont Res. 2019 Jul 11. pii: S1883-1958(18)30435-3.; Dobrzynski et. al.: Study of Surface Structure Changes for Selected Ceramics Used in the CAD/CAM System on the Degree of Microbial Colonization, In Vitro Tests. Biomed Res Int. 2019; 2019: 9130806.), as well as disadvantages metal-ceramic restorations.

Author Response

Reply:Thank you very much for your comments

 Our crowns are made with CAD/CAM techniques and I agree with you and considered important to include some references of articles of this issue.

Introduction:

Line 58-61.

Monolithics materials are made with CAD/CAM techniques that requires only few steps and less time for the fabrication of a restoration compared to old traditional methodsbut researches have noted in some cases worse marginal fit and a lower longevity.

We included the main disadvantages of metal-ceramic material:

Line 40-41.

the disadvantages in its esthetic appearance and biocompatibility.

We modified the discussion include some sentences to clarify:

Monolithic zirconia crowns are made with CAD/CAM techniques and allow more conservative dental preparation [13,16,17]. In addition, the thickness allows us to obtain higher transclucency values and a decreased contrast ratio to improve a long-term aesthetic compared with metal-ceramic[18].

We have modified the conclusions to adapt them to the results obtained

Yttrium-stabilized tetragonal zirconia is the only other material used to fabricate full coverage crowns that equaled the fracture resistance obtained by the metal-ceramic control group.Following these two materials in decreasing order of fracture resistance values were: zirconia core with feldspathic ceramic covering, PMMA doped with graphene nanoparticles, and yttrium-stabilized cubic zirconia. Within the monolithic zirconia crowns tested, tetragonal zirconia presented higher fracture resistance than cubic zirconia. All the crowns analyzed, internal core and monolithic restorations, obtain adequate load resistance values for correct oral behavior. Prospective medium and long-term clinical studies are needed to support the survival of restorations made with CAD/CAM monolithic materials. 

Round 2

Reviewer 1 Report

thank You for complying to the suggestion